# Is Working from Home during COVID-19 Associated with Increased Sports Participation? Contexts of Sports, Sports Location and Socioeconomic Inequality

**DOI:** 10.3390/ijerph191610027

**Published:** 2022-08-14

**Authors:** Malou Grubben, Sara Wiertsema, Remco Hoekman, Gerbert Kraaykamp

**Affiliations:** 1Department of Sociology, Radboud University, P.O. Box 9104, 6500 HE Nijmegen, The Netherlands; 2Mulier Institute, P.O. Box 85445, 3508 AK Utrecht, The Netherlands

**Keywords:** remote working, health behavior, sports participation, corona crisis

## Abstract

Previous research has focused mainly on the association between working from home (WFH) and physical activity, establishing that physical activity diminished among people WFH during the COVID-19 pandemic. In our study, we investigated the association between WFH and specifically sports participation (competitive and non-competitive). We theorized that WFH would offer individuals additional opportunities to practice sports during the pandemic. Governmental restrictions at the time constrained opportunities to participate in organized sports and in sports with others. We, therefore, expected sports participation during the pandemic to be largely restricted to individual participation and participation at home or in the public space. By means of descriptive analyses and adjusted analyses of variance (*n* = 1506), we found positive associations between WFH and various aspects of sports participation. Lower-educated individuals, in particular, seem to be benefiting from WFH related to their sports participation in the public space, and economically deprived individuals also seem to be benefiting from WFH in regard to their sports participation at home. Our findings extend the literature on physical activity and sports participation among people who worked from home during the COVID-19 pandemic while offering implications for policies on WFH, sports opportunities in public space and physical activity-friendly environments.

## 1. Introduction

The World Health Organization classified COVID-19 as a global pandemic on 12 March 2020. Shortly thereafter, the Dutch government issued several public health mandates to reduce the spread of the virus, such as social distancing, self-quarantining and closure of recreational facilities [1]. This disrupted people’s daily routines and, for many, led to increased stress, anxiety, insomnia and loneliness [2,3,4]. For those in the labor force who could work from home, working from home (WFH) during the pandemic became obligatory. For some members of the workforce, their jobs precluded working from home. Besides essential jobs (e.g., in care, education and security), some jobs require specialized machinery or tasks carried out at a particular location (e.g., factories, construction sites, hospitals and retail). In prior empirical studies, WFH has been associated with less social interaction, more work uncertainties, sedentary behavior, unhealthy eating and blurred work–home boundaries [5,6,7,8].

Sports participation and physical activity may help to alleviate such health problems. Being sufficiently physically active during the COVID-19 pandemic has been associated in prior research with less loneliness, anxiety, stress and insomnia [9,10,11,12]. Sports participation benefits individuals’ physical health as well [13,14]; and a lack of sports participation has even been associated with a higher risk of serious complications following a COVID-19 infection [15]. This illustrates the importance of maintaining existing sports participation and promoting sports participation and other physical activity among inactive people.

Concerning physical activity, Xiao et al. [16] found a relation between WFH and lower physical activity, and other studies have produced similar associations [6,17,18]. Nevertheless, there are also signals that WFH may actually be conducive to sports participation, due to greater task flexibility, better work–life balance, increased work productivity and reduced commuting time [19]. Indeed, sports participation, both competitive and non-competitive (e.g., joining a spinning class or sport club training sessions), requires more time and planning than other forms of physical activity that may be more integrated into daily life (e.g., walking to the supermarket). Sports participation may therefore have been differently impacted by WFH during the COVID-19 pandemic than other forms of physical activity. The measure of sports participation that we employ is a subjective self-estimate of what the respondent perceives as being sports participation. This entails both competitive and non-competitive sports activities (e.g., running, fitness activities, competitive club sports). 

WFH can allow workers to devote more time to sports, as they may spend less time commuting and be stimulated to work more efficiently [20,21]. Additionally, WFH may give workers more control over working hours, enabling them to take breaks when they prefer and be more flexible with starting and finishing times [7,22,23]. WFH therefore may make it easier for workers to practice sports during breaks and/or adapt their working routines to facilitate sports participation, compared to workers who have to work on location. Furthermore, WFH may offer individuals more flexibility to adapt their sports participation to the COVID-19 restrictions. For example, to slow the spread of the virus, the Dutch government mandated the closure of sports clubs and grassroots sports facilities (either completely or during the evening hours); it also banned sporting with others [1]. This severely restricted opportunity for sports participation among the Dutch population [24]. People could still, however, participate in sports individually and at home or in the public space. We, therefore, expected those WFH to be more active in sports during the COVID-19 pandemic, especially in individual types of sports and sports that can be practiced at home (e.g., fitness activities and stationary cycling) or in the public space (e.g., running). Much previous research on WFH does not distinguish between the contexts or locations of sports [16,18]. This study extends previous research by distinguishing individual sports participation and participation at home or in the public space. With this distinction, we aim to provide a more complete picture of the role of WFH in enabling sports participation in various contexts and locations.

As observed by previous studies, the COVID-19 pandemic has magnified existing inequalities in sports participation [25,26,27]. In our study, we therefore pay specific attention to differences in sports participation between socioeconomic status (SES) groups in regard to WFH. A reason for increasing inequalities due to the COVID-19 pandemic is that lower socioeconomic groups have fewer resources and experience more restrictions in adapting to the COVID-19 measures. First, even though WFH applies to workers in both low and high socioeconomic positions, the autonomy of workers in lower socioeconomic positions is more restricted [28,29]. This may manifest as strict time schedules or intensive monitoring of work tasks. Consequently, such workers cannot reap the benefits of WFH (i.e., flexibility), and this may hamper their sports participation relative to that of their higher socioeconomic WFH counterparts.

Second, practicing a sport at home and/or in the public space requires the opportunity to do so. People with less favorable socioeconomic positions often live in smaller or more crowded homes and less favorable neighborhoods [30,31]. Thus, while lower SES workers may have the desire and time to participate in sports when they are WFH, they likely have fewer opportunities to do so. Whereas most previous studies do control for the socioeconomic position of workers [16,17], we explicitly study the differential effects of WFH on sports participation for the various SES groups.

Overall, our research questions are: To what extent did WFH during the COVID-19 pandemic affect sports participation, specifically individual participation, and participation at home or in the public space? Additionally, to what extent are these relationships different for lower and higher SES groups?

We tested our expectations using high-quality panel data from the Longitudinal Internet Studies for the Social Sciences (LISS). In so doing, our aim was to provide a more complete picture of the impact of WFH during the COVID-19 pandemic on various aspects of sporting behavior. This knowledge can be utilized for better policies to organize and manage WFH in the future, and to stimulate people WFH to participate in sports. Our findings show positive associations between WFH and the various measures of sports participation. Among lower SES groups who worked from home, we found that the lower educated in particular seem to benefit from WFH regarding their sports participation in the public space, while the economically deprived particularly seem to benefit in regard to their sports participation at home and in the public space.

## 2. Materials and Methods

### 2.1. Data and Respondents

To answer our research questions, we collected data between 7 June and 27 July 2021 within the nationwide Dutch LISS panel (for more information about the LISS panel, see www.lissdata.nl (accessed on 17 August 2021)). This panel consists of a representative random sample of approximately 5000 households drawn from Dutch population registers. These households comprised approximately 7500 individuals aged 16 and older. Panel members were questioned monthly on different topics, with core modules repeated each year. To study the impact of WFH on sports participation, we merged the LISS module in which we elicited elaborate information on sports participation in March, April and May of 2021, with general information from the LISS core module on work (collected in April and May 2021). Higher educated respondents were slightly overrepresented in LISS. We therefore weighted our data on educational level, sex and age group to match the distribution of these characteristics in the Netherlands in 2021.

The weighted sample consisted of respondents in paid employment for at least eight contracted hours a week and with valid answers to the questions on sports (*n* = 1538). Respondents were considered working when they indicated being in paid employment, working or assisting in a family business, or being an autonomous professional, freelancer or self-employed as their primary activity. This sample of workers consisted of individuals aged 18 and older. We excluded respondents with missing information on variables such as educational level, economic deprivation and relevant controls (~2%). Our final sample consisted of 1506 working respondents (Table 1). Women and men were almost equally represented, with 51% of respondents being female and 49% male. Respondents were divided into three age groups: 18–34 years (22%), 35–54 years (47%) and 55+ years (31%). Concerning educational attainment, 47% were higher educated, 38% had a middle-level education and 15% had a lower educational level. Note that this high representation of higher educated reflects a general characteristic of Dutch workers, who in general hold higher educational attainment than non-workers. The distribution of education in our sample is close to the distribution of educational level in the Dutch labour force reported by Statistics Netherlands (2021).

### 2.2. Measurements

In our study, we focused on the impact of WFH on various measures of sports participation. To avoid confusion, we explicitly list the operationalizations of the different sports measures below.

We asked respondents if they had participated in sports in the last three months (March, April and May 2021). In the LISS questionnaire, respondents were free to report on sports participation as they understood it, so the interpretation of sports was subjective. Respondents were classified into two categories, namely not participating in sport and participating in sport.

Sports participants were also asked whether they had participated in a sport individually. Respondents who indicated they did so at least once a month were given a score, while others were not scored.

We also asked sports participants where they had participated in sports. Among the possible locations were at home and in the public space. Those indicating they had participated at these locations at least once a month were given a score for the sports location concerned.

As noted, our study selected respondents who were considered to be working based on their primary daily activity. They were asked how many hours they spent WFH in a normal week during COVID-19 (April and May 2021). Based on this information, we categorized respondents who worked from home for at least one hour as WFH and the others as not WFH.

Respondents indicated their educational attainment by the standard Dutch qualifications, for which we used the corresponding ISCED levels. We divided the respondents into three groups based on their highest completed level of education: lower educated (intermediate secondary education or lower), middle educated (higher secondary education and intermediate vocational education) and higher educated (higher vocational education or university).

We additionally asked respondents six questions related to financial difficulties experienced in March, April and May 2021. These explored whether respondents had a hard time making ends meet, were unable to replace broken belongings, received money from friends or family, borrowed money for necessary expenditures, were behind in paying rent/mortgage and had a debt collector at their door. Respondents could answer ‘yes’ or ‘no’ to these questions. If they answered yes on one of the items, we coded them as economically deprived.

In the analyses, we controlled for several factors that might confound the relationship between WFH and sports participation outcomes: sex (female/male), age group (18–34, 35–54, 55+) and weekly contracted working hours (8–20, 21–34, and 35+ h). Additionally, we retrieved whether respondents had children and, if so, the age of those children from the LISS Family and Household core module. Respondents were coded into two groups: having no children or having children older than 12 years of age versus having at least one child aged 12 or younger. Concerning COVID-19, we included whether health complaints related to a COVID-19 infection hampered the extent to which a respondent could participate in sports (yes/no). Finally, we controlled for whether respondents agreed with the statement that sufficient sports provisions were available in their neighborhood (yes/no). In Table 1, descriptive statistics of the variables are presented.

**Table 1 ijerph-19-10027-t001:** Descriptives, *N* = 1506.

	Minimum	Maximum	Mean (%)	Std. Dev
**Dependent variables**				
Sports participation (ref. = no)	0	1	47	0.499
Individual participation (ref. = no)	0	1	35	0.478
Participation at home (ref. = no)	0	1	14	0.349
Participation in public space (ref. = no)	0	1	35	0.479
**Independent variables**				
Working from home (ref. = no)	0	1	49	0.500
Educational level				
Lower educated	0	1	15	0.352
Middle educated	0	1	37	0.484
Higher educated	0	1	48	0.500
Economic deprivation (ref. = no)	0	1	8	0.268
Male (ref. = female)	0	1	49	0.500
Age group				
18–34 years	0	1	21	0.406
35–54 years	0	1	46	0.499
55+ years	0	1	33	0.468
Having a child ≤ 12 (ref. = no)	0	1	21	0.412
Hampered by COVID-19 (ref. = no)	0	1	11	0.313
Weekly working hours				
8–20 h	0	1	13	0.324
21–34 h	0	1	30	0.462
35+ h	0	1	57	0.495
Sufficient sports provisions (ref. = no)	0	1	83	0.373

### 2.3. Analytical Strategy

We conducted our analyses using the statistical software package SPSS v27. First, we determined the means and Pearson correlations of all outcome variables and predictors (Table 1 and Table 2). Analyses of variance (ANOVA), which were adjusted by all covariates, were then performed to test the influence of WFH on several outcomes relating to sports participation. Table 3 reports the adjusted scores from multiple classification analyses (MCA) within ANOVA. Reported scores may be seen as predicted means of the subsequent groups. In other words, MCA means represent the predicted percentage of, for example, respondents WFH that participated in sports. This analysis technique was preferred as it provided immediately interpretable output that clearly showed deviations from the grand mean for specific social categories (in Table 1). The eta squared in Table 3 is a measure of effect size based on the proportion of variance of a variable.We descriptively explored moderation by educational attainment and financial situation, presented in Figure 1 and Figure 2. To test whether the effect of WFH on sports participation significantly differed between the SES groups, we conducted significance testing with ANOVA. Table 4 presents MCA scores for the groups separately.

**Table 2 ijerph-19-10027-t002:** Pearson correlations.

Variable	1		2		3		4		5		6		7		8		9	
**1** Sports participation	1.000																	
**2** Individual participation	0.778	***	1.000															
**3** Participation at home	0.427	***	0.455	***	1.000													
**4** Participation in public space	0.778	***	0.821	***	0.309	***	1.000											
**5** Working from home	0.185	***	0.194	***	0.099	***	0.194	***	1.000									
**6** Educational level: lower	−0.179	***	−0.162	***	−0.117	***	−0.166	***	−0.250	***	1.000							
**7** Educational level: middle	−0.135	***	−0.101	***	−0.076	**	−0.125	***	−0.163	***	1.000		1.000					
**8** Educational level: higher	0.258	***	0.212	***	0.156	***	0.239	***	0.334	***	1.000		1.000		1.000			
**9** Economic deprivation	−0.076	**	−0.067	**	−0.033		−0.067	*	−0.043		0.081	**	0.016		−0.073	**	1.000	
Male	−0.031		−0.004		−0.101	***	0.014		0.014		0.078	**	0.017		−0.072	**	−0.018	
Age group: 18–34	0.098	***	0.050	+	0.082	**	0.028		0.026		−0.143	***	−0.090	**	0.188	***	0.005	
Age group: 35–54	−0.038		−0.036		−0.016		−0.032		−0.012		−0.019		0.074	**	−0.058	*	0.015	
Age group: 55+	−0.045	+	−0.006		−0.054	*	0.010		−0.009		0.145	***	−0.001		−0.102	***	−0.020	
Having a child ≤ 12 (ref. = no)	−0.021		−0.007		−0.029		−0.019		0.073	**	−0.074	**	−0.014		0.066	*	0.022	
Hampered by COVID-19	0.022		0.004		−0.010		−0.013		−0.043	+	0.002		0.088	**	−0.087	**	0.044	+
Weekly working hours: 8–20	−0.019		−0.036		−0.001		−0.048	+	−0.116	***	0.114	***	0.022		−0.102	***	0.048	+
Weekly working hours: 21–34	−0.035		−0.014		−0.006		−0.015		−0.078	**	−0.063	*	0.034		0.011		0.038	
Weekly working hours: 35+	0.046	+	0.037		0.006		0.046	+	0.149	***	−0.019		−0.047	+	0.059	*	−0.068	**
Sufficient sports provisions	0.106	***	0.083	**	0.007		0.083	**	0.036		−0.071	**	−0.030		0.079	**	−0.093	***

*N* = 1506. + *p* < 0.10; * *p* < 0.05; ** *p* < 0.01; *** *p* < 0.001 (two tailed).

**Table 3 ijerph-19-10027-t003:** Adjusted predicted mean MCA estimates in percentages and adjusted ANOVA significance levels.

		Sports Participation			Context of Participation			Location of Participation					
					Individual ^a^			At Home ^a^			Public Space ^a^		
		Mean		Eta^sq^	Mean		Eta^sq^	Mean		Eta^sq^	Mean		Eta^sq^
Working from home	Yes	53	***	0.185	42	***	0.194	16	***	0.099	41	***	0.194
	No	42			29			12			30		
Educational level	Lower	30	***	0.275	20	***	0.232	6	***	0.176	19	***	0.252
	Middle	40			30			11			29		
	Higher	59			44			19			46		
Economic deprivation	Yes	41	+	0.069	29	+	0.065	11		0.040	30		0.065
	No	48			36			14			36		
Sex	Female	48		0.029	35		0.002	17	**	0.097	34		0.015
	Male	47			35			11			37		
Age group	18–34	51	**	0.096	36		0.048	16	**	0.084	34		0.005
	35–54	47			35			15			35		
	55+	45			36			12			37		
Having a child ≤ 12	Yes	43		0.015	33		0.004	11		0.027	32		0.005
	No	49			36			15			37		
Hampered by COVID-19	Yes	54		0.022	38		0.000	14		0.011	38		0.011
	No	47			35			14			35		
Weekly working hours	8–20	49		0.056	35		0.053	15		0.010	34	*	0.073
	21–34	44			35			12			35		
	35+	49			36			15			36		
Sufficient sports provisions in neighborhood	Yes	49	***	0.104	37	**	0.087	14		0.018	37	**	0.101
	No	38			28			14			29		

*N* = 1506. + *p* < 0.10; * *p* < 0.05; ** *p* < 0.01; *** *p* < 0.001 (two tailed). ^a^ Comparison: Respondents who participated in sports individually/at home/in the public space vs. respondents who did not participate in sports individually/at home/in the public space and those that did not participate in sports at all.

**Table 4 ijerph-19-10027-t004:** Adjusted predicted mean MCA estimates and adjusted ANOVA outcomes of the interaction effects.

	Sports Participation		Context of Participation		Location of Participation			
			Individual ^a^		At Home ^a^		Public Space ^a^	
	Means		Means		Means		Means	
*Educational attainment*	WFH	Not WFH	WFH	Not WFH	WFH	Not WFH	WFH	Not WFH
Lower	28	24	25	14	7	3	18	15
Middle	45	35	38	24	12	9	35	24
Higher	65	53	50	39	21	17	51	39
*Economically deprived*								
Yes	42	31	29	22	12	7	34	18
No	54	43	43	30	16	13	42	31

*N* = 1506. The interactions were not found to be significant. Note: All estimates were adjusted by the variables in Table 3. ^a^ Comparison: Respondents who participated in sports individually/at home/in the public space vs. respondents who did not participate in sports individually/at home/in the public space and those that did not participate in sports at all.

**Figure 1 ijerph-19-10027-f001:**
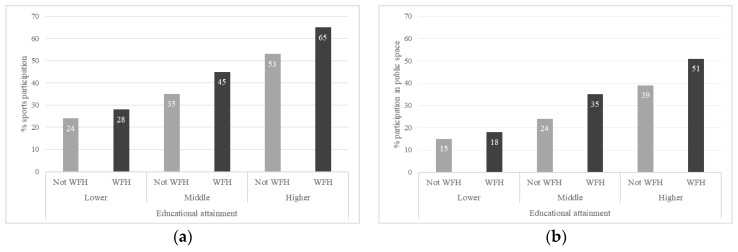
Visualization of the relationship between WFH and sporting behavior, differentiated by educational level for (**a**) sports participation and (**b**) participation in the public space.

## 3. Results

### 3.1. Descriptive Information

Table 1 provides a description of the variables we employ in the ANOVA analyses. Almost half of our sample worked from home (49%). Concerning sports, 47% participated in sports and 35% participated individually. Sports participation in the public space (35%) was more prevalent than participating at home (14%). Eight percent of our sample was economically deprived. Table 2 presents correlations among the variables, we see that WFH and sports participation were positively correlated (*r* = 0.185, *p* = 0.000). We also observe a positive association between WFH and individual sports participation (*r* = 0.194, *p* = 0.000), sports participation in the public space (*r* = 0.194, *p* = 0.000) and sports participation at home (*r* = 0.099, *p* = 0.000). These correlations suggest that individuals who worked from home indeed engaged more in sports, individually, at home and in the public space, than individuals who did not work from home.

### 3.2. ANOVA Analyses

#### 3.2.1. Sports Participation, Context and Location

Concerning sports participation, Table 3 shows significant differences between workers who did and did not work from home, adjusted for relevant characteristics. More than half of those who worked from home participated in sports (53%), whereas only 42% of the workers who did not work from home participated in sports. This implies a substantial positive relationship between WFH and sports participation during the COVID-19 pandemic.

We also examined whether there were substantial differences between people who did and did not work from home in the context and location of their sports participation. Regarding context, 42% of those who worked from home participated in sports individually, compared to 29% of the workers who did not work from home (Table 3). This significant difference indicates a greater prevalence of individual sports participation among workers who worked from home. 

With regard to location, participating in sports at home was more common among workers who worked from home (16%) compared to workers who did not work from home (12%) (see Table 3). Moreover, participating in sports in the public space was more prevalent among workers who worked from home (41%), compared to workers who did not work from home (30%). These significant differences imply that a relatively large number of people who worked from home during the COVID-19 pandemic participated in sports at home and in the public space.

With respect to the control variables, these show that sports participation during the COVID-19 pandemic was higher among non-financially deprived, higher educated and younger people, and people with sufficient sports provisions in their neighborhood. Concerning the context of sports, higher educated and non-financially deprived workers and those perceiving sufficient sports provisions in their neighborhood were more likely to participate in sports individually. Regarding the location of sports, a relatively large number of women, higher educated and young people participated in sports at home, whereas sports participation in the public space was relatively more common among the higher educated, those who worked more hours and those who lived in neighborhoods with sufficient sports provisions.

#### 3.2.2. Differences between SES Groups: Moderation by Educational Attainment and Economic Deprivation

Descriptive Figure 1 and Figure 2 display mean scores on some of the dependent variables for workers who did and did not work from home, differentiated by educational level and financial situation. The figures only display outcomes where the socioeconomic factors seemed to affect the relationship between WFH and sports participation. The figures show differences in means of WFH on sports participation for educational attainment and financial hardship categories. Figure 1a,b suggest that the positive association of WFH with sports participation and sports participation in the public space was stronger for the higher educated than for the middle and lower educated. Figure 2a indicates that the positive relation between WFH and participating in sports individually was stronger for those who were not economically deprived, whereas Figure 2b,c suggest that the positive relation of WFH and sports participation at home and in the public space was stronger among economically deprived workers compared to those who were not economically deprived.

**Figure 2 ijerph-19-10027-f002:**
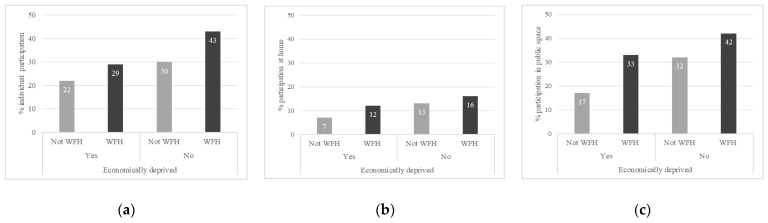
Visualization of the relationship between WFH and sporting behavior, differentiated by economic deprivation for (**a**) individual sports participation, (**b**) participation at home and (**c**) participation in the public space.

Table 4 investigates the significance of the moderations observed in the descriptive figures. Unfortunately, the estimates show no significant moderation in the association for educational level or financial situation, neither for sports participation nor for the context of sports or the location of sports. This indicates that workers’ educational level and financial situation did not significantly affect the relationship between WFH and sporting behavior. The fact that neither moderator reached significance likely was due to our small subsamples of lower educated and economically deprived workers, rather than a complete lack of moderation.

### 3.3. Additional Analyses

WFH may have differently affected the sports participation of men and women, as working women may assume greater care responsibilities than working men [32]. We therefore performed the analyses separately for men and women (see Appendix A). In general, we concluded that WFH positively impacted the sports participation of both sexes. In other words, men and women were similarly affected by WFH concerning their sports participation, individual participation, and participation at home or in the public space.

Our main analyses measured WFH as remote working at least one hour a week. To test robustness, we examined a linear operationalization of WFH to observe whether an increase in WFH hours was related to an increased chance of sports participation (see Appendix A). This analysis showed that more WFH indeed increased the chance of participating in sports, participating individually and participating at home or in the public space.

## 4. Discussion

The findings of the current study offer new knowledge on the relationship between WFH and sports participation during the COVID-19 pandemic. Previous research has focused mainly on the link between WFH and physical activity [5,6,17,18], establishing that physical activity diminished among those who worked from home during the pandemic. Our focus here was on the association between WFH and specifically sports participation, as we expected that WFH would provide additional sports opportunities, especially during the COVID-19 pandemic, when governmental restrictions limited organized sports.

Our findings indicate a positive association between WFH and sports participation, as the prevalence of sports participation in general, individually, at home and in the public space was higher among those who worked from home compared to those who did not. These results align with a review study by Tavares [19], which found that flexibility and autonomy gave workers more opportunities to participate in sports. Furthermore, Yang et al. [33] found that, in a Chinese population, people who worked from home for longer than three weeks were more likely to change their sporting behavior compared to people who worked from home for one week only. Taken together, these findings suggest that WFH can be a stimulus for sports participation.

Contradictive results have been found, however, particularly in studies reporting on the link between WFH and physical activity [6,16,17,18]. For example, Argus and Pääsuke [18] found a negative correlation between WFH and physical activity in Estonia during the COVID-19 pandemic. Other studies on the association between WFH and physical activity reached similar conclusions [6,16,17]. In our study, the focus was on being active in sports and its association with WFH during the COVID-19 pandemic. Our study indicates that WFH affected sports participation differently than it, according to aforementioned studies, affects physical activity: in contrast to physical activity, WFH was positively associated with sports participation.

A vast body of research demonstrates that there is a socioeconomic gap in sports participation, which is explained in many ways [34,35]. Although we found no substantial moderation effects, we did find preliminary evidence that, in regard to the relationship between WFH and sports participation, higher-educated individuals benefited more from WFH than workers with a lower education. This aligns with several studies concluding that the educational inequality in sports participation increased during the COVID-19 pandemic [24,25,26]. The more common practice of WFH during the pandemic may have contributed to magnifying the educational inequality in sports participation as higher educated seemed to benefit more in their sports participation from WFH. Moreover, our findings thus seem to align with the expectation that educational level is positively correlated with flexibility and autonomy in work [36,37]. Elldér [38], for example, found that lower status jobs offered less control over scheduling when WFH compared to higher status jobs. So it seems likely that in our sample too, workers’ sporting behavior was substantially affected by the autonomy and flexibility of their job when they worked from home. For employers, this might be a signal to allow remote workers more leeway to take breaks or to be flexible in setting working times, in order to stimulate workers’ health and fitness.

A recent Dutch study on the association between WFH and vigorous physical activity found a dichotomy among people who worked from home: some were more physically active during the COVID-19 pandemic, whereas others became less vigorously physically active [39]. Our results similarly point to differing effects of WFH on sporting behavior, particularly by educational level. We encourage future research to explore this moderation more elaborately both theoretically and empirically. For example, lower SES workers may benefit more from WFH when given more autonomy and when living in an environment conducive to an active lifestyle. At present, lower SES groups are known to be less physically active across Europe [40]. At the same time, neighborhoods with a higher representation of lower SES individuals tend to be less physical activity friendly (e.g., due to safety concerns, walkability, and lack of parks and trails [26,41,42]). This mismatch between the additional sports opportunities of WFH and the objective environment seems an exciting puzzle for sports stimulation policy makers to solve.

Our outcomes thus yield implications for both employers and policy makers seeking to promote sports participation. Though we conducted our study during the COVID-19 pandemic, and we were mainly interested in the impact of WFH on contexts and locations of sports participation that were significantly affected by the COVID-19 restrictions, our results may nonetheless be relevant after the pandemic. Several studies pre-dating COVID-19 also suggest that the flexibility and autonomy that come with WFH bring more opportunities to participate in sports [7,20,21,22,23]. Consequently, there seems to be an argument for organizations to encourage workers to work from home under the right conditions, even after the COVID-19 pandemic. Particularly, remote workers should be given sufficient autonomy and flexibility to maximize the opportunities that WFH offers for sports participation. Additionally, we encourage policy makers seeking to stimulate sports participation to pay special attention to providing physical activity-friendly public spaces in neighborhoods. Our findings, among others, demonstrate that people who worked from home participated relatively often individually in sports in the public space. This might especially hold for economically disadvantaged individuals WFH. Thus, to combat the social inequality in sports participation, facilitating physical activity-friendly public spaces in more deprived neighborhoods seems a key challenge. Indeed, as our correlations matrix shows, lower educated and economically deprived workers agreed less with the statement that their neighborhood had sufficient sports provisions.

Our study has limitations as well. First, we tested our hypotheses using a Dutch sample. This specificity may have affected our outcomes since the Netherlands has a relatively high level of organized sports participation [43]. The fact that governmental mandates restricted participation in organized sports for a long period during the pandemic may have affected more people than in other countries. Nonetheless, we believe our results regarding WFH are generalizable to workers outside the Netherlands because of the general idea of WFH corresponding with more autonomy and flexibility. Second, our relatively small sample seems to have hindered full statistical testing of the moderation influences of educational attainment and economic deprivation. A larger sample of working individuals would be needed to confirm our indicative conclusions. Finally, issues of reverse causality or selectivity are often mentioned as a drawback in studies of sports participation. Our approach was characterized by a sports causation perspective (i.e., individuals’ work experiences affect subsequent sports participation). A reverse causal direction, however, is also imaginable: sports participation and a subsequent better health situation might lead to more WFH. Future research therefore should seek opportunities to perform longitudinal analyses on the relationship between WFH and sports participation.

## 5. Conclusions

In 2020, government mandates to slow the spread of COVID-19 forced people to work from home. This study examined the extent to which WFH brought additional opportunities for sports participation during the pandemic (in March, April and May 2021). Controlling for several confounding factors, we found that WFH was positively associated with sports participation: the level of sports participation was relatively high among workers who worked from home compared to those who did not. This positive association of WFH and sports participation was most prominent among higher educated workers. However, lower educated remote workers particularly benefited in regard to their sports participation in the public space; while economically deprived individuals particularly benefited from WFH in regard to their sports participation at home and in the public space. Policy makers might therefore pay special attention to developing flexibility and autonomy among lower SES workers who work from home, creating physical activity-friendly public space, particularly in less advantaged neighborhoods, to ensure that everyone has the opportunity to actively engage in sports.

## Data Availability

Restrictions apply to the availability of these data. Data was obtained from LISS and are available https://www.lissdata.nl, accessed on 11 August 2022 with the permission of LISS.

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
