# Peer review of "Is Working from Home during COVID-19 Associated with Increased Sports Participation? Contexts of Sports, Sports Location and Socioeconomic Inequality"

_ijerph, 2022, doi:10.3390/ijerph191610027_

Round 1
Reviewer 1 Report
Abstract:If you want to use the acronym WFH you must put in the first occurrence of the expression work from home (WFH) with curly brackets instead of straight brackets. If you intend physical activity to include leisure time, this expression should appear explicitly and not in parentheses. Do not use references in the abstract. Please remove from the abstract "we examined 1506 Dutch workers (LISS, 2021)". It uses the acronym SES without explaining its meaning beforehand. Same with ANOVA despite it being a widely known acronym.
Introdution: References should appear in straight brackets and not in curly brackets. For example on line 32: [1]. In lines 39 and 40 delete the parentheses from the words health, sufficiently and risk. In line 97 why does moderation appear in parentheses? I think the sentence in lines 103-107 should not be in the Introduction.
Materials and Methods: The acronym LISS should be explained only once. There should be no unnumbered topics, such as: Sports participation (line 137). Being short paragraphs, they should not have titles. In the methodology you should briefly explain what an ANOVA analysis is. Explain MCA. Tables cited in the text should appear immediately after their citation. In Figure 1 the letters (a) and (b) appear below the subfigures, and in Figure 2 they appear above. In Figure 2 the figures do not maintain their original proportion. Tables and Figures must have the source.
Discussion: Why does the word appear in parentheses in the line 318? Lines 340-370 should appear in the conclusions. The discussion should be improved by taking into account results from similar studies.
Reviewer 2 Report
Dear Authors
Thank you for your submission. Although my review of your work entails a rework of the statistical approach followed, I would like to encourage you to resubmit the manuscript once addressed. As always, feel free to offer a rebuttal if you disagree with my review.
Overall, I missed a clear distinction between "sport" and exercise. What sports would one practice for instance at home, individually? For instance, some consider chess and competitive eSports to be a sport.
ABSTRACT:
Line 20 - 23 "Moreover, we ..." this sentence should be rephrased. In it's current form, it is unclear what you would like to say. Also do not copy and paste from the text (and vice versa) as this sentence is similarly structured in lines 104-107.
INTRODUCTION:
Line 95: Why does the research question specifically state a positive affect? Should it not be neutral in the research question, with you formulating an answer without any bias?
MATERIALS AND METHODS:
You state that LISS included individuals older than 16, however, there is no mention of those individuals in your work. According to a quick internet search, individuals older than 16 are capable of entering full-time employment in the Netherlands. https://business.gov.nl/regulation/employment-young-people/#:~:text=Children%20aged%20between%20the%20ages,receive%20the%20minimum%20youth%20wage.
You exclude those individuals that do not have a full set of answers. With education being one of the primary measurements, I wondered whether you introduced a bias in your data by doing so. It might be that those less educated were less successful in completing the questionnaire. If this is the case, then your study design selects more individuals with a higher education level, which in turn results in a higher representation of those WFH. It would be interesting to know whether your results on educational attainment is similar to that of the full LISS dataset.
Under measurements, you have subheadings, which stop at "Locations of sports participation". The next paragraphs do not have similar headings.
You coded those respondents that had children into three groups. However, in subsequent analyses only one group is reported. Why?
Table 1 should be presented in Results, not in materials and methods.
What does MCA refer to? It has not been correctly abbreviated at first presentation.
You do not mention how correlation was done. Spearman? Pearson? Kendall Tau?
RESULTS
The results throughout should be reworked extensively. Results should be written in past tense - this was the case in 2021 and it may no longer be relevant.
You present the Means. Although not entirely incorrect, one should be wary of calculating the mean of a nominal variable. I would rather present the Means in table 1 as a percentage of the study population.
Paragraph 3.1 you state that positive correlations were observed (r = .185) Please note that the correct way of presenting these results is with a 0 preceeding the decimal (r = 0.185). Also note that a correlation of 0.185 is considered a very weak correlation. I would not make any assumptions on a low correlation, even with a significant p-value.
On the topic of p-values, please keep your writing consistent. In some instances you refer to P and in others p. This should be as per the guidelines of the journal (NEJM uses P, and as far as I know IJERPH uses p-value).
Subsequent analyses are problematic. For instance, in Table 2 you indicate Means - which are impossible if you used your dataset. These should rather be % in my opinion. Also problematic is the fact that these do not add up to 100. Your educational level is split into three groups, with a total of 129%; unless I am interpreting the results incorrectly. This expands throughout the rest of the results, and I would appreciate more clarity on how you analysed your data.
I look forward to your rebuttal and wish you the very best with this manuscript.
Kind regards
Reviewer 3 Report
Please see the attached file.

Reviewer 4 Report
I believe that the fact of basing most of the results on correlations does not allow certain conclusions to be affirmed, since correlations are not cause-effect relationships. They show associations between variables but they are not a consequence of each other. In addition, the degree of association has not been described, which, on the other hand, we can say is low. On the other hand, the previous sports experience of the participants has not been described. This may be a conditioning factor since despite the restrictive measures during the pandemic, those who have never practiced physical activity do not even consider it in that situation. In table 1 where the sample is described, I think it would be more correct to put the frequencies and not the mean to have a global idea. On the other hand, in the comparative graphs it does not appear if the differences are significant or not. Finally, I think that the study is not so related to the effect of the restrictions in a pandemic situation, or at least it might not be. Since it is not known if in a normal teleworking situation (without a pandemic) these results would be the same or not. Therefore, more than the pandemic, perhaps it would be necessary to analyze what other factors may be conditioning the practice of physical activity.
Round 2
Reviewer 2 Report
Dear Authors
Thank you for your resubmission. The points were well addressed.
I thought that "oefening" and "sport" would be considered different constructs by Dutch speakers, but since I am not Dutch (but Afrikaans), I'll abide by your interpretation. Although I understand that "sports" may include both participatory competition and exercise in Dutch, the paper is presented in an English journal. All the available definitions for "sport" involve a competitive aspect as well. As such, I still have reservations on the use of "sport" in this article. I would rather use the term physical activity - which would include both competitive sport, as well as personal exercise.
Author Response
Dear reviewer,
You will find our response in the attachment.

Reviewer 4 Report
See the attach document

Author Response

(The authors gave the same response as above.)
